# Planning with Consistency Models for Model-Based Offline Reinforcement Learning

**Guanquan Wang**                                                    *guanquan-wang@g.ecc.u-tokyo.ac.jp*
*Department of Information and Communication Engineering*
*The University of Tokyo*

**Takuya Hiraoka**                                                              *takuya-h1@nec.com*
*NEC Corporation, Tokyo, Japan*

**Yoshimasa Tsuruoka**                                            *yoshimasa-tsuruoka@g.ecc.u-tokyo.ac.jp*
*Department of Information and Communication Engineering*
*The University of Tokyo*

**Reviewed on OpenReview:** *https://openreview.net/forum?id=TuACCzfty3*

## Abstract

This paper introduces consistency models to the problem of sequential decision-making. Previous work applying diffusion models to planning within a model-based reinforcement learning framework often struggles with high computational cost during the inference process, primarily due to their reliance on iterative reverse diffusion processes. Consistency models, known for their computational efficiency, have already shown promise in reinforcement learning within the actor-critic algorithm. Therefore, we combine guided consistency distillation with a continuous-time diffusion model in the framework of Decision Diffuser. Our approach, named Consistency Planning, combines the robust planning capabilities of diffusion models with the speed of consistency models. We validate our method on Gym tasks in the D4RL framework, demonstrating that, when compared to its diffusion model counterparts, our method achieves more than a 12-fold increase in speed without any loss in performance.

## 1 Introduction

In recent years, significant strides have been made in high-resolution image generation through the advancement of diffusion-based generative models. Similarly, in offline reinforcement learning (RL) settings, deriving effective policies from pre-existing offline datasets can be simplified to the task of developing a probabilistic model for trajectory prediction, an area where diffusion-based generative models have proven to be highly successful. Existing models such as Diffuser (Janner et al., 2022) and Decision Diffuser (Ajay et al., 2022) underscore the efficacy of applying diffusion models to planning within model-based RL frameworks. In Diffuser, a diffusion model is trained on the trajectories in offline datasets, and then a separate classifier model is trained to predict the cumulative rewards of trajectory samples. During the inference process, the diffusion model, combined with classifier guidance, is employed to sample trajectories with high returns. Likewise, Decision Diffuser introduces a conditional diffusion model with state sequences as input, utilizing the return as a conditioning variable for classifier-free guidance during sampling. Moreover, by incorporating only the state sequence—excluding the action sequence—Decision Diffuser trains an extra inverse dynamic model to infer actions.

Parallel to these developments, diffusion models have been adapted to model-free reinforcement learning scenarios, as illustrated by Diffusion-QL (Wang et al., 2022) and further enhanced in the efficient diffusion policy (EDP) (Kang et al., 2024). Diffusion-QL, utilizing a denoising diffusion probabilistic model (DDPM)

(Ho et al., 2020), frames the diffusion model as a policy representation, conditioned on states with actions as outputs. It integrates Q-learning guidance into the reverse diffusion process to seek optimal actions. Despite its advancements, Diffusion-QL faces limitations in computational efficiency and its exclusive application within TD3-type algorithms (Fujimoto & Gu, 2021). EDP addresses these issues by introducing an action approximation trick during training, applying the DPM-solver, and approximating policy likelihood via the evidence lower bound in the DDPM to overcome the limitations of Diffusion-QL.

Although the integration of diffusion models within both model-based and model-free frameworks in the offline RL setting has been extensively explored and enhanced, a significant challenge remains in their application, particularly in real-time decision-making contexts. This challenge stems from the diffusion models' reliance on iterative sampling processes, which can be computationally intensive and slow, thus restricting their use in scenarios that require rapid inference. For instance, in robot arm control (Chi et al., 2023), standard diffusion-based control can only make decisions at around 10Hz. However, this is insufficient for tasks requiring agile motion planning at 20Hz (Smith et al., 2023), 30Hz (Peng et al., 2020), or even higher.

Recent endeavors by Song et al. (2023) have introduced consistency models, a novel class of generative models that significantly enhance computational efficiency without sacrificing the expressiveness and flexibility that make diffusion models appealing for reinforcement learning.

In the model-free RL domain, consistency models have demonstrated promising results as policy representations, particularly in offline and offline-to-online RL settings (Ding & Jin, 2023). These developments underscore consistency models' capability to effectively navigate the challenges of learning from fixed datasets, indicating their potential to achieve performance comparable to diffusion-based approaches, while maintaining higher computational efficiency.

However, in offline RL settings, model-free methods using Q-network face challenges due to overestimated Q-values for out-of-distribution actions (Kumar et al., 2020; Levine et al., 2020). In the context of online RL, the problem is self-correcting as the policy interacts with the environment; an action perceived as favorable might receive a low reward, thus adjusting the policy. However, in offline RL, such corrections are not readily achievable, which often leads the learned Q-function to often guide the diffusion model towards potentially sub-optimal actions. Therefore, given the computational efficiency of consistency models and the proven effectiveness of diffusion models in trajectory prediction, this paper aims to explore how consistency models can augment model-based RL with classifier-free guidance in an offline setting, thereby bypassing the necessity of learning a Q-function by conditioning the consistency models on returns.

The goal of this paper is to bridge this gap by proposing a novel approach that merges the computational efficiency of consistency models with the planning capabilities inherent in Decision Diffuser. By integrating consistency models into the trajectory optimization process, we aim to leverage their computational advantages to enhance the speed of planning. Our experiments, conducted in offline RL settings, embed a conditional consistency model in the Decision Diffuser algorithm, evaluating with consistency distillation methods. Specifically, the Consistency Model employs guided consistency distillation from a score-based diffusion model (Karras et al., 2022; Ho & Salimans, 2022; Luo et al., 2023) pretrained on offline trajectory datasets.

In summary, our contribution is proposing Consistency Planning, a novel offline RL algorithm that extends the applicability of consistency models to model-based RL. We evaluate Consistency Planning on D4RL benchmark tasks (Fu et al., 2020) for offline RL, demonstrating that this method can achieve performance comparable to its diffusion model counterparts across the majority of tasks, while offering a notably faster sampling process.

## 2 Related Work

### 2.1 Diffusion Models

Diffusion models have emerged as a powerful approach for generating high-quality image and text data, as demonstrated by previous studies (Saharia et al., 2022; Nichol & Dhariwal, 2021). The data sampling

process is formulated as an iterative denoising procedure, introduced by Sohl-Dickstein et al. (2015) and further developed by Ho et al. (2020). Parameterizing the gradients of the data distribution serves as an alternative interpretation of this denoising procedure, aiming to optimize the score matching objective, as elucidated by Hyvärinen & Dayan (2005). This positions the approach within the domain of Energy-Based Models, as evidenced by the contributions of Du & Mordatch (2019), Nijkamp et al. (2019), and Grathwohl et al. (2020).

Prior work (Nichol & Dhariwal, 2021) has implemented a classifier to enable the generation of images based on conditional information (e.g., text), which is called classifier guidance. However, more recent studies (Ho & Salimans, 2022), propose classifier-free guidance, which relies on the gradients from an implicit classifier, derived from the score function differences between conditional and unconditional models. This approach has proven to enhance the quality of conditional samples compared to classifier guidance methods. These advancements predominantly focus on text and image generation.

## 2.2 Diffusion Models in Reinforcement Learning

Diffusion models offer a versatile approach for data augmentation in reinforcement learning. SynthER (Lu et al., 2024) employs unguided diffusion models to enhance both offline and online RL datasets, subsequently utilized by model-free off-policy algorithms. Although this approach boosts performance, SynthER's reliance on unguided diffusion to approximate the behavior distribution faces challenges due to distributional shift. Similarly, MTDiff (He et al., 2024) implements unguided data generation in multitask environments.

Additionally, diffusion models have been adapted for training world models. For instance, Alonso et al. (2023) use diffusion to train world models, achieving precise predictions of future observations. However, this method does not model entire trajectories, leading to compounded errors and lack of policy guidance. In a related effort, Rigter et al. (2023) integrate policy guidance to enhance a diffusion world model in online RL. Jackson et al. (2024) concentrate on offline RL, providing a theoretical framework and rationale for the trajectory distribution shaped by policy guidance.

Diffusion models have also been adapted for policy representation in RL, capturing the multi-modal distributions in offline datasets. Specifically, Diffusion-QL (Wang et al., 2022), applies the diffusion model within the framework of both Q-learning and Behavior Cloning (BC) for policy representation. However, the main limitation of Diffusion-QL is that it demonstrates computational inefficiency due to the necessity of processing both forward and backward through the entire Markov chain during training. To alleviate these issues, Kang et al. (2024) introduce action approximation, eliminating the need to execute the denoising process during the training process.

Diffusion models have been employed in recent studies for human behavior imitation learning (Pearce et al., 2023) and trajectory generation in offline RL. Trajectories that include states and actions are generated by Diffuser (Janner et al., 2022), using an unconditional diffusion model, guided by a reward function trained on noisy state-action pairs. Decision Diffuser (Ajay et al., 2022) models the trajectories with the dataset using a unified, conditional generative model, avoiding separate training a classifier for reward functions.

# 3 Preliminary

## 3.1 Reinforcement Learning Problem Setting

The sequential decision-making problem is defined as a Markov decision process (MDP): $M = \{S, A, P, R, \gamma, d_0\}$, where $S$ and $A$ are the state space and the action space respectively, $P : S \times A \rightarrow S$ represents the transition function, $R : S \times A \times S \rightarrow \mathbb{R}$ denotes the reward function, $\gamma \in [0, 1)$ is the discount factor, $d_0$ is the initial state distribution. The goal of RL is to learn policy $\pi_\theta (a|s)$ to maximize the expected sum of discounted rewards $\mathbb{E}\left[\sum_{k=0}^{\infty} \gamma^k r(s_k, a_k)\right]$.

## 3.2 Consistency Models

Diffusion models operate by introducing Gaussian perturbations to transform data into noise, followed by generating data samples through a series of sequential denoising steps. Song et al. (2020) introduce a stochastic differential equation (SDE) framework that ensures the maintenance of the desired distribution as sample $x$ evolves over time. The consistency models proposed by Song et al. (2023) recover the original data sample by solving a corresponding probability flow ordinary differential equation (ODE): $\frac{\mathrm{d}x_t}{\mathrm{d}t} = -t\nabla \log_t p_t(x)$, where $p_t(x) = p_{data}(x) \otimes \mathcal{N}(0, t^2\mathbf{I})$, $p_{data}(x)$ represents the original data distribution, $t \in [0, T]$ is the time period. The data generation process in this framework reverses along the trajectory $\{\hat{x}_t\}_{t\in[\epsilon,T]}$ of the ODE, starting from random initial samples $\hat{x}_T \sim \mathcal{N}(0, T^2\mathbf{I})$ where $\epsilon$ is a minimal constant close to 0 to address numerical issues at the boundary.

To accelerate the sampling process in diffusion models, the consistency model significantly reduces the number of steps required for sampling compared to the original diffusion model, without substantially compromising the model's performance. This is achieved by approximating a parameterized consistency function, $f_\theta : (x_t, t) \to x_\epsilon$, which maps a noisy sample $x_t$ at step $t$ back to the original sample $x_\epsilon$.

This approach differs from the diffusion model, which utilizes a step-by-step denoising function $p_\theta(x_{t-1} \mid x_t)$, for the reverse diffusion process. Slightly different from the original consistency model, this paper focuses on a conditional distribution, so the consistency function is modified to $f_\theta(x_t, t, c)$, where $c$ denotes the condition variable.

# 4 Planning with Consistency Model

This paper explores the integration of consistency models, which are trained by distillation from a pre-trained diffusion model, into the planning architecture of Decision Diffuser. The original consistency models draw from the principles of score-based diffusion models (Song et al., 2020; Karras et al., 2022), making direct distillation from the discrete-time model used in Decision Diffuser ineffective. In the following, we discuss how we use consistency models for the trajectory optimization process. Section 4.1 details the training process of the diffusion model, followed by an explanation in Section 4.2 of how guided consistency distillation is applied, and how the consistency model is integrated with the Decision Diffuser framework during inference.

## 4.1 Diffusion Model Training

**Trajectory representation.** As outlined by Ajay et al. (2022), the diffusion process encompasses only the state transitions as described by

$$x_{t_{i+1}}(\tau) := (s_k, s_{k+1}, \ldots, s_{k+H-1})_{t_{i+1}}. \tag{1}$$

In this notation, $k$ indicates the timestep of a state within a trajectory $\tau$, $H$ represents the planning horizon, and $t_{i+1}$ is the timestep in the diffusion sequence. Consequently, $x_{t_{i+1}}(\tau)$ is defined as a noisy sequence of states, represented as a two-dimensional array where each column corresponds to a different timestep of the trajectory. In training process, the sub-sequence $t_i \in [\epsilon, T]$ follows the Karras boundary schedule (Karras et al., 2022):

$$t_i = \left(\epsilon^{1/\rho} + \frac{i-1}{N-1}\left(t_N^{1/\rho} - \epsilon^{1/\rho}\right)\right)^\rho, \tag{2}$$

where $\epsilon = 0.002$, $t_N = 80$, and $\rho = 7$.

**Acting with inverse dynamics.** To derive actions from the states generated by the diffusion model, we employ an inverse dynamics model (Agrawal et al., 2016; Pathak et al., 2018), denoted as $h_\varphi$, trained using the same dataset as the diffusion model. Actions can be obtained via the inverse dynamics model by extracting the consecutive state $s_k$ and $s_{k+1}$ at diffusion timestep $t_0$:

$$a_k = h_\varphi(s_k, s_{k+1}). \tag{3}$$

**Combined training of diffusion models and inverse dynamics.** Given a dataset $\mathcal{D}$ consisting of trajectories, each labeled with its respective return, the combined training of the diffusion model (denoted by $D_\phi$) and the inverse dynamics model is conducted using the following loss:

$$
\begin{aligned}
\mathcal{L}(\phi, \varphi) :=& \mathbb{E}_{\sigma \sim p_{train}, \tau \sim \mathcal{D}, n \sim \mathcal{N}(0, \sigma^2 \mathrm{I}), \beta \sim \mathrm{Bern}(p)} \left[ \|D_\phi(x_\sigma(\tau), (1 - \beta)c(\tau) + \beta\emptyset, \sigma) - x_0(\tau)\|_2^2 \right] \\
& + \mathbb{E}_{(s, a, s') \sim \mathcal{D}} \left[ \left\| a - h_\varphi(s, s') \right\|_2^2 \right],
\end{aligned}
\tag{4}
$$

where $p_{train}$ is a log-normal distribution using the design choice from Karras et al. (2022), $\beta$ is sampled from a Bernoulli distribution with probability $p$. Namely, the condition information $c(\tau)$ is ignored with probability $p$, which is manifested by the condition information being an empty set $\emptyset$.

We employ returns $R(\tau)$ under trajectories as the conditioning information $c(\tau)$, normalized such that $R(\tau) \in [0, 1]$. We map it into a latent variable $c \in \mathbb{R}^h$ using a multi-layer perceptron. In cases where $R(\tau) = \emptyset$, the components of c are set to zero. During the inference time, sampling trajectories with high returns corresponds to conditioning on $R(\tau) = 1$.

## 4.2 Consistency Distillation

**Guided Consistency Distillation.** Incorporating classifier-free guidance is essential for synthesizing high-return trajectories. Considering the computational demands and potential for error accumulation associated with two-stage distillation methods (Meng et al., 2023), we opt for a one-stage guided distillation approach as proposed by Luo et al. (2023).

---

**Algorithm 1** Consistency Distillation with guidance

1: **Input**: dataset $\mathcal{D}$, intial consistency model parameter $\theta$, learning rate $\eta$, ODE solver $\Phi(\cdot, \cdot, \cdot; \phi)$, distance metric $d(\cdot, \cdot)$, EMA rate $\mu$, noise schedule $t_i$, guidance schedule $[\omega_{min}, \omega_{max}]$.
2: Initialize target consistency model $\theta^- \leftarrow \theta$
3: **repeat**
4:     Sample trajectory and condition $(x, c) \sim \mathcal{D}$, $n \sim \mathcal{U}[1, N-1]$ and guidance $\omega \sim \mathcal{U}[\omega_{min}, \omega_{max}]$
5:     Sample noised trajectory $x_{t_{n+1}} \sim \mathcal{N}(x; t_{n+1}^2 \mathrm{I})$
6:     $\hat{x}_{t_n}^{\phi, \omega} \leftarrow x_{t_{n+1}} + [(\omega+1)\Phi(x_{t_{n+1}}, c, t_{n+1}; \phi) - \omega\Phi(x_{t_{n+1}}, \emptyset, t_{n+1}; \phi)]$     // Classifier-free guidance
7:     $\mathcal{L}(\theta, \theta^-; \phi) \leftarrow d\left( f_\theta(x_{t_{n+1}}, \omega, c, t_{n+1}), f_{\theta^-}(\hat{x}_{t_n}^{\phi, \omega}, \omega, c, t_n) \right)$     // Calculate consistency distillation loss
8:     $\theta \leftarrow \theta - \eta\nabla_\theta \mathcal{L}(\theta, \theta^-; \phi)$     // Update consistency model parameter
9:     $\theta^- \leftarrow \mathrm{stopgrad}(\mu\theta^- + (1-\mu)\theta)$     // Update target consistency model parameter
10: **until** convergence

---

The consistency function $f_\theta : (x_t, \omega, c, t) \to x_0$ is parameterized to transform state $x_t$ at time $t$ directly into the original state $x_0$. We parameterize $f_\theta$ in the same way as Song et al. (2023), except that we consider the influences of guidance scale $\omega$ and conditioning variable $c$:

$$
f_\theta(x, \omega, c, t) = c_{skip}(t)x + c_{out}(t)F_\theta(x, \omega, c, t),
\tag{5}
$$

where $F_\theta$ is a free-form neural network with an output that matches the dimensionality of $x$, $c_{skip}(\epsilon) = 1$ and $c_{out}(\epsilon) = 0$ so that $f_\theta$ satisfies boundary condition $f_\theta(x, \omega, c, \epsilon) \equiv x$. During the distillation process, the guidance scale $\omega$ and $n$ are sampled uniformly from the intervals $[\omega_{min}, \omega_{max}]$ and $\{1, \cdots, N-1\}$, respectively. The trajectory and returns tuple $(x, c)$ are sampled from the dataset. Then, $\hat{x}_{t_n}^{\phi, \omega}$ is estimated by employing an ODE solver $\Phi$:

$$
\hat{x}_{t_n}^{\phi, \omega} - x_{t_{n+1}} \approx \left[ (\omega+1)\Phi(x_{t_{n+1}}, c, t_{n+1}; \phi) - \omega\Phi(x_{t_{n+1}}, \emptyset, t_{n+1}; \phi) \right].
\tag{6}
$$

Finally, we minimize the consistency distillation loss (Song et al., 2023; Luo et al., 2023) used for guided distillation:

$$\mathcal{L}(\theta, \theta^-; \phi) = \mathbb{E}_{x,c,\omega,n} \left[ d \left( f_\theta(x_{t_{n+1}}, \omega, c, t_{n+1}), f_{\theta^-}(\hat{x}_{t_n}^{\phi,\omega}, \omega, c, t_n) \right) \right], \tag{7}$$

where $d$ is squared $\ell_2$ distance $d(x, y) = \|(x - y)\|_2^2$.

The pseudo-code for guided consistency distillation adapted for trajectory generation is shown in Algorithm 1.

---

**Algorithm 2** Planning with Consistency Model

---

1: **Input**: consistency model $f_\theta$, inverse dynamics $h_\varphi$, guidance scale $\omega$, history length $C$, condition $c$, sequence of time points $t_1 > t_2 > ... > t_{N-1}$, initial noise $x_T$, fixed small positive number $\epsilon$.
2: Initialize history $h \leftarrow Queue(length = C)$, $t \leftarrow 0$.    // Keep a history with a maximum length C
3: **while** not *done* **do**
4:     Observe state $s$; $h.insert(s)$;
5:     Initialize $x(\tau) \leftarrow f_\theta(x_T, \omega, c, T), x_T \sim \mathcal{N}(0, T^2\mathrm{I})$   // Generate samples and evaluate the consistency model in a single step
6:     **for** $n = 1$ to $N - 1$ **do**                    // Evaluate the consistency model in multiple steps
7:         $x(\tau)[:, length(h)] \leftarrow h$                    // Ensure the trajectory aligns with the history
8:         $\hat{x}_{t_n}(\tau) \leftarrow x(\tau) + \sqrt{t_n^2 - \epsilon^2} z, z \sim \mathcal{N}(0, \mathrm{I})$
9:         $x(\tau) \leftarrow f_\theta(\hat{x}_{t_n}(\tau), \omega, c, t_n)$
10:    **end for**
11:    Extract state tuple $(s_k, s_{k+1})$ from generated trajectory $x(\tau)$
12:    Execute action $a_k = h_\varphi(s_k, s_{k+1})$
13: **end while**

---

**Consistency Model Inference.** During the inference process, we first observe a state $s$ in the environment and sample an initial trajectory $x_T$. Then, our consistency model, conditioned on returns $c$, guidance scale $\omega$ and history of last C states observed, iteratively predicts the denoised trajectories from the noisy inputs $\hat{x}_{t_n}(\tau) \leftarrow x(\tau) + \sqrt{t_n^2 - \epsilon^2} z$ along the probability flow ODE trajectory at step $n \in [N]$, with Gaussian noise $z \sim \mathcal{N}(0, \mathrm{I})$. For the single-step version of Consistency Inference, $\{t_n \mid n = 0, 1\} = \{\epsilon, T\}$. Finally, we extract states $(s_k, s_{k+1})$ from denoised trajectory and get the action $a_k$ via our inverse dynamics model $h_\varphi$. The algorithm of Consistency Planning is provided in Algorithm 2 and visualized in Figure 1. For the architecture and implementation details, please refer to Appendix.

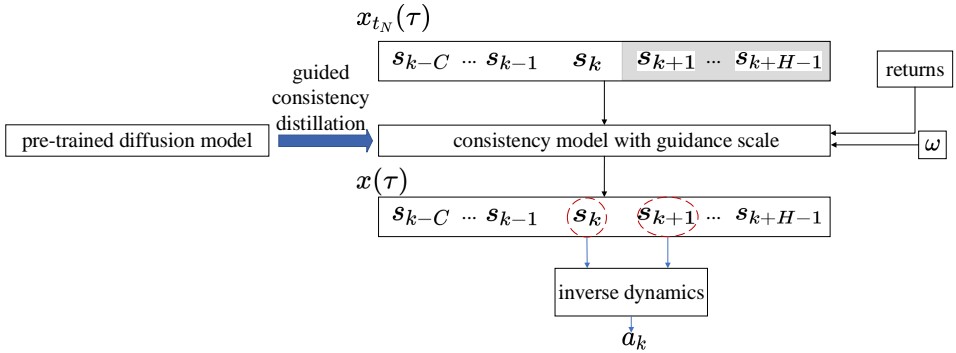

Figure 1: Consistency Planning. Given the current state $s_k$, conditioning variable and guidance scale $\omega$, Consistency Planning generate a sequence of future states with planning horizon $H$. Then the inverse dynamics model is used to extract and execute the action $a_k$ from $s_k$ and $s_{k+1}$

# 5 Experiment

In our experiment section, we evaluate the performance of the Consistency Planning on offline RL settings and its computational efficacy in Section 5.1, and provide a detailed ablation study to test the effectiveness of the inverse dynamic model $h_\varphi$ and the guidance scale $\omega$ in Section 5.2.

## 5.1 Offline Reinforcement Learning

The diffusion model, inverse dynamics model, and consistency model are trained using publicly available D4RL datasets and subsequently evaluated on a range of Gym tasks, including HalfCheetah, Hopper, Walker2d, and Maze2d, within the D4RL benchmark suite (Fu et al., 2020). These tasks are characterized by continuous state and action spaces and are conducted under offline reinforcement learning settings. Details of the dataset size are provided in Appendix.

We compare the performance of our method with those of both behavior-cloning methods, i.e., Consistency-BC (C-BC) (Ding & Jin, 2023), Diffusion-BC (D-BC) (Wang et al., 2022), actor-critic methods, i.e., Consistency-AC (C-AC) (Ding & Jin, 2023), Diffusion-QL (D-QL) (Wang et al., 2022) algorithms, and model-based methods, i.e., Diffuser (Janner et al., 2022), Decision-Diffuser (DD) (Ajay et al., 2022) in Table 1. For evaluation, results for our method correspond to the average over 150 planning seeds. By default, our consistency model applies the number of denoising steps N = 2 with saturated performance on most tasks, while the diffusion policy uses $N = 5$ (Wang et al., 2022), Diffuser and Decision Diffuser use $N = 20$ and $N = 40$, respectively (Janner et al., 2022; Ajay et al., 2022). For the performance metric, we use normalized average returns (Fu et al., 2020) for all tasks.

Table 1 shows that although our method achieves a slightly lower average score (82.2) than Diffusion-QL (87.9) and Consistency-AC (85.1), as a model-based planning model, it outperforms its diffusion counterparts, i.e., Diffuser (75.3) and Decision Diffuser (81.8), with the reduction of denoising steps in the inference stages.

However, it is worth noting that our model exhibits a more pronounced performance gap in the half-cheetah environment compared to the hopper and walker2d environments compared with Diffusion-QL and Consistency-AC. This can be attributed to several factors. While our model employs classifier-free guidance, it lacks the value function estimation provided by Q-networks in Consistency-AC and Diffusion-QL. These Q-network-based methods explicitly optimize for cumulative rewards, allowing them to perform better in environments like half-cheetah, where subtle variations in action sequences can have a significant impact on long-term returns. Moreover, half-cheetah presents a higher degree of control complexity and a more dynamic action space compared to hopper and walker2d. The need for precise coordination among multiple joints makes it more challenging for generative models like ours, which rely purely on the consistency model and classifier-free guidance, to generate optimal action sequences. In contrast, Q-network-based methods are better equipped to navigate this complexity by explicitly evaluating state-action pairs in terms of their future rewards.

To validate the long-horizon planning capabilities of Consistency Planning, we conduct an evaluation in the Maze2d environment (Fu et al., 2020), where the task involves navigating to a specific goal location, with a reward of 1 assigned only upon reaching the goal. Since it requires hundreds of steps to reach the goal, even the most advanced model-free algorithms struggle with effective credit assignment and consistently reaching the goal. As shown in Table 2, Consistency Planning achieved scores exceeding 100 across all maze sizes, indicating that it outperforms the reference expert policy.

To assess the computational efficiency of Consistency Planning and its diffusion model counterparts, Decision Diffuser, we conduct experiments to measure inference time (ms per sample) in the hopper-medium-expert-v2 on our server. Both the consistency model and the diffusion model, as generative models based on probability flow, have computational times that are directly dependent on the number of denoising steps $N$. The results in Figure 2 show that $N = 2$ for Consistency-Planning, and $N = 20$ for Decision Diffuser, are the values where each algorithm achieves its saturated performance in hopper-medium-expert. The mean and standard deviation of results are calculated over five random seeds.

Table 1: The average scores of vanilla BC (with Gaussian), Diffuser, Decision Diffuser, Diffusion-BC, Consistency-BC, Diffusion-QL, Consistency-AC and our method on D4RL Gym tasks are shown, with standard deviation reported for Consistency Planning. All results are quoted from Ding & Jin (2023) and Ajay et al. (2022).

| Dataset | BC | Diffuser | DD | D-BC | C-BC | D-QL | C-AC | Ours |
|---|---|---|---|---|---|---|---|---|
| Halfcheetah-me | 55.2 | 79.8 | 90.6 | 90.8 | 32.7 | **96.8** | 84.3 | $94.0 \pm 1.3$ |
| Hopper-me | 52.5 | 107.2 | **111.8** | 107.6 | 90.6 | 111.1 | 100.4 | $107.5 \pm 1.8$ |
| Walker2d-me | 107.5 | 108.4 | 108.8 | 108.9 | **110.4** | 110.1 | **110.4** | **109.8**±0.5 |
| Halfcheetah-m | 42.6 | 44.2 | 49.1 | 45.4 | 31.0 | 51.1 | **69.1** | $46.8 \pm 1.2$ |
| Hopper-m | 52.9 | 58.5 | 79.3 | 65.3 | 71.7 | **90.5** | 80.7 | $87.8 \pm 1.6$ |
| Walker-m | 75.3 | 79.7 | 82.5 | 81.2 | 83.1 | **87.0** | 83.1 | $80.5 \pm 0.8$ |
| Halfcheetah-mr | 36.6 | 42.2 | 39.3 | 41.7 | 34.4 | 47.8 | **58.7** | $40.6 \pm 0.9$ |
| Hopper-mr | 18.1 | 96.8 | 100 | 67.9 | 99.7 | **101.3** | 99.7 | $97.8 \pm 0.8$ |
| Walker2d-mr | 26.0 | 61.2 | 75 | 77.5 | 73.3 | **95.5** | 79.5 | $75.3 \pm 1.1$ |
| Average | 51.9 | 75.3 | 81.8 | 76.3 | 69.7 | **87.9** | 85.1 | 82.2 |

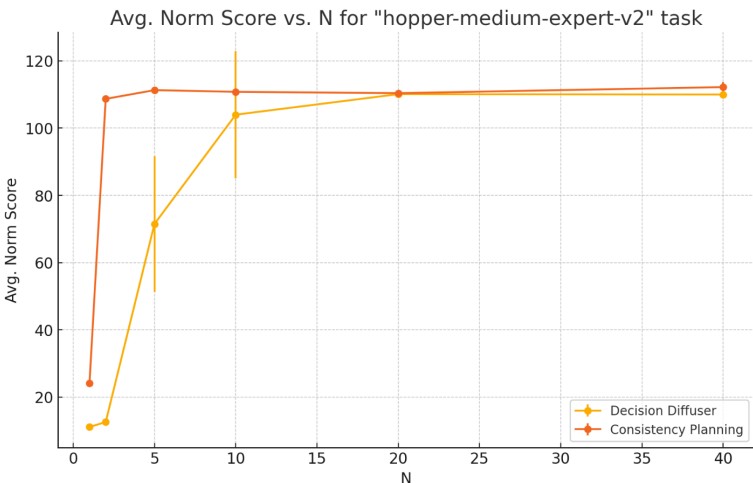

Figure 2: Comparison of average norm score vs. N for Decision Diffuser and Consistency Planning on the task hopper-medium-expert-v2

Since the computational time for these models is directly tied to $N$, and the consistency model is designed to require fewer denoising steps to achieve similar generative performance, we claim our model has achieved more than a 12-fold increase in speed without any loss in performance. More detailed information concerned the

Table 2: The performance of Consistency Planning, Diffuser, and previous model-free algorithms in the Maze2D environment, which tests long-horizon planning due to its sparse reward structure. Consistency Planning's performance is comparable to that of Diffuser, with diffusion steps $N = 2$. All results are derived from the data provided in Janner et al. (2022).

| Dataset | MPPI | CQL | IQL | Diffuser | Ours |
|---------|------|-----|-----|----------|------|
| Maze2D U-Maze | 33.2 | 5.7 | 47.4 | $113.9 \pm 3.1$ | $\mathbf{122.7}\pm2.7$ |
| Maze2D Medium | 10.2 | 5.0 | 34.9 | $\mathbf{121.5}\pm2.7$ | $\mathbf{121.4}\pm4.1$ |
| Maze2D Large | 5.1 | 12.5 | 58.6 | $\mathbf{123.0}\pm6.4$ | $119.5 \pm 7.5$ |
| Average | 16.2 | 7.7 | 47.0 | 119.5 | $\mathbf{121.2}$ |

Table 3: Ablation study on inverse dynamics.

| Hopper-* | Consistency Planning w/o inverse dynamics | Ours |
|----------|-------------------------------------------|------|
| Med-Expert | $105.9 \pm 1.7$ | $110.4 \pm 2.4$ |
| Medium | $78.3 \pm 2.4$ | $88.7 \pm 2.2$ |
| Med-Replay | $89.4 \pm 3.8$ | $96.7 \pm 1.6$ |

inference time v.s. $N$ across the other hopper environments, halfcheetah-medium-expert-v2 and walker2d-medium-expert-v2 shown in Appendix.

## 5.2 Ablation Study

**Inverse dynamics.** To evaluate the role of inverse dynamics in our approach, we conducted a comparative analysis with Consistency Planning models that exclude the inverse dynamics component, where the trajectories consist of both state and action sequences. The performance results are presented in Table 3. Our findings indicate that incorporating inverse dynamics leads to better performance across all three hopper environments, consistently outperforming the variant without inverse dynamics. This demonstrates the effectiveness of inverse dynamics in improving the model's ability to generate action sequences, contributing enhanced planning performance.

**Guidance Scale.** In our experiments, we examine the impact of guidance scale $\omega$ on the performance of our consistency models during planning. The guidance scale controls the weight of the guidance signal during the planning process, and its effect on performance can vary significantly depending on the characteristics of the offline dataset. To assess the sensitivity of our model to $\omega$, we conducted experiments across different types of datasets, including hopper medium-expert, hopper medium, and hopper medium-replay.

For the hopper medium-expert dataset, our results in Figure 3 indicate that varying the guidance scale $\omega$ does not lead to substantial differences in performance. This can be attributed to the relatively high-quality data in the medium-expert dataset. As a result, the model performs well across a wide range of $\omega$ values, and the influence of the guidance signal is less pronounced.

However, when applied to hopper medium and medium-replay datasets, we observe a marked difference in performance based on the choice of $\omega$. Specifically, larger guidance scales (e.g., $\omega = 0.8$) consistently result in better outcomes compared to smaller scales (e.g., $\omega = 0$). This is particularly evident in the medium setting, where the dataset contains suboptimal trajectories alongside more desirable ones. In such cases, the stronger guidance provided by larger $\omega$ helps the model better navigate the diverse quality of trajectories, ultimately leading to more consistent and improved performance. On the other hand, smaller $\omega$ values may not provide sufficient signal strength to guide the model away from suboptimal decisions, resulting in degraded performance.

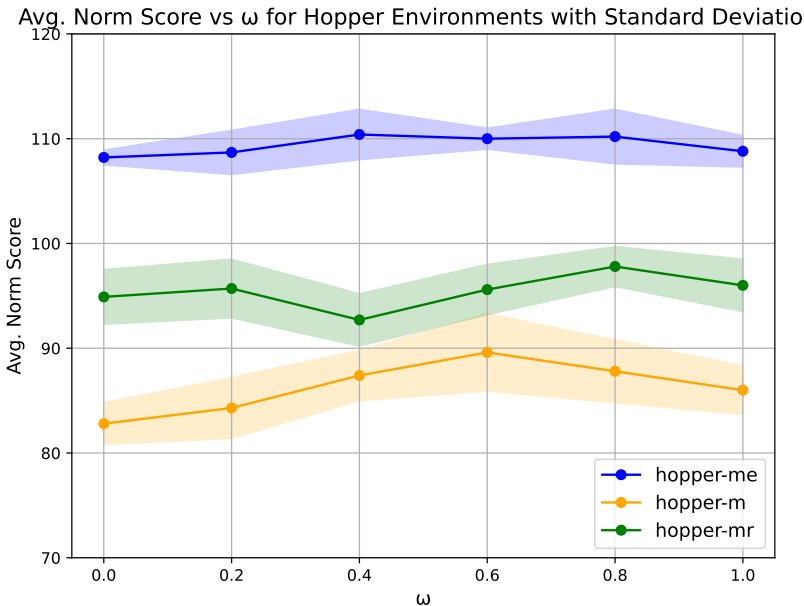

Figure 3: Ablation study on different guidance scales $\omega$ across 3 hopper environments. Larger $\omega$ leads to better sample quality.

## 6 Conclusion

By combining the score-based diffusion model proposed by Karras et al. (2022), one-stage guided distillation (Luo et al., 2023), and conditional model-based generative model for sequential decision making (Ajay et al., 2022), the consistency model in this paper achieves comparable performance in gym tasks with its diffusion model counterparts, Diffuser and Decision Diffuser, and obtains a significant speedup during inference in offline settings. While our work increases the inference speed compared with its diffusion models counterpart (Ajay et al., 2022), there still exists some potential challenges of applying the approach to scenarios where quick, real-time decision making is critical.

One key challenge is scalability and model complexity. As consistency models are applied to larger and more complex environments, such as autonomous driving or stock market trading, the number of variables and possible future states grows exponentially. Managing this complexity in real-time, while ensuring consistent and reliable outcomes, presents a significant challenge without the use of advanced computational techniques. Another challenge involves data availability. Effective real-time decision-making requires access to up-to-date and accurate data streams. In an online setting, the Consistency Planning method requires completing an entire episode to collect the trajectory before updating the model, which can lead to suboptimal results in dynamic environments. This data collection process introduces delays that may hinder the effectiveness of real-time applications.

Future work should include: 1) combining improved techniques in training consistency models (Song & Dhariwal, 2023), such as designing a changing weighting function and noise schedule more suitable for reinforcement learning scenarios; 2) combining the consistency inference process with changing guidance schedule (Ma et al., 2023) to improve the quality of trajectory sampling; and 3) investigating online learning strategies to reduce delays and improve performance in real-time scenarios.

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

# A   Appendix

Table 4 is a full list of dataset size. We use the latest version of dataset in D4RL (Fu et al., 2020). Our experiments utilize D4RL datasets comprising three distinct data types for the Hopper, HalfCheetah, and Walker environments: medium-expert, medium, and medium-replay. The medium-expert datasets contain trajectories generated through a combination of medium-level and expert-level policies, offering a blend of both optimal and suboptimal actions. The medium datasets are produced exclusively by medium-level policies and therefore exhibit a greater proportion of suboptimal actions compared to the medium-expert datasets. The medium-replay datasets represent the replay buffer of a medium-level agent, encompassing a diverse range of suboptimal actions along with exploration noise.

The results in Table 5 show the relationship between the computational time, denoised steps $N$ and corresponding performance of Consistency Planning and Decision Diffuser on the task hopper-medium-expert-v2. The results in Table 6 show the inference time and corresponding performance of Consistency Planning and Decision Diffusuer in different datasets, i.e, hopper-mr, hopper-m, halfcheetah-me and walker2d-me, with diffusion steps $N = 100$ for Decision Diffuser and $N = 2$ for Consistency Planning. The choice of diffusion steps in Table 6 depends on the default training hyperparamters in Ajay et al. (2022) which claims all tasks suffer no performance loss with $N = 100$. Each cell contains the mean and standard deviation over 5 random seeds in the same server. As demonstrated in Table 5, we achieved more than 12-fold increase in speed without any loss in performance with N=2 for Consisitency Planning and N=20 for Decision Diffuser.

Table 4: Size for each dataset is provided. The number of samples indicates the total count of environment transitions recorded in the dataset (Fu et al., 2020).

| Dataset | Samples |
|---|---|
| Hopper-me | $2 \times 10^6$ |
| Hopper-m | $10^6$ |
| Hopper-mr | 402000 |
| Halfcheetah-me | $2 \times 10^6$ |
| Halfcheetah-m | $10^6$ |
| Halfcheetah-mr | 202000 |
| Walker-me | $2 \times 10^6$ |
| Walker-m | $10^6$ |
| Walker-mr | 302000 |
| Maze2D U-Maze | $10^6$ |
| Maze2D Medium | $2 \times 10^6$ |
| Maze2D Large | $4 \times 10^6$ |

In the next section, we describe various architectural and hyper-parameter details:

- We parameterize the diffusion model and consistency model with a temporal U-Net architecture, a neural network consisting of repeated convolutional residual blocks from Janner et al. (2022), with 2nd order Heun as ODE solver, the inverse dynamics $h_\varphi$ using the structure of Ajay et al. (2022).

- We train diffusion model using learning rate of $1e-4$ and batch size of 512 for $2e5$ train steps with Adam optimizer.

- We choose the probability $p$ of removing the conditioning information to be 0.25.

- We use $N = 2$ for consistency inference.

- We use a planning horizon H of 32, context length C of 8 in all tasks.

- We use a guidance scale $\omega_{max} = 1, \omega_{min} = 0$ in guided consistency distillation.

Table 5: Comparison of computational time for Decision Diffuser and Consistency Planning on the task hopper-medium-expert-v2 (Ajay et al., 2022).

| Method | N | Inference Time (ms per sample) | Avg. Norm Score |
|---|---|---|---|
| Decision Diffuser | 40 | $837.6 \pm 8.4$ | $110.0 \pm 0.4$ |
| | 20 | $427.7 \pm 3.9$ | $110.1 \pm 0.5$ |
| | 10 | $216.8 \pm 1.2$ | $104.0 \pm 18.9$ |
| | 5 | $107.3 \pm 0.5$ | $71.5 \pm 20.2$ |
| | 2 | $44.5 \pm 0.3$ | $12.6 \pm 0.5$ |
| | 1 | $23.1 \pm 0.4$ | $11.1 \pm 0.5$ |
| Consistency Planning | 40 | $752.1 \pm 2.0$ | $112.2 \pm 1.5$ |
| | 20 | $331.9 \pm 1.9$ | $110.4 \pm 0.3$ |
| | 10 | $167.3 \pm 0.9$ | $110.8 \pm 1.0$ |
| | 5 | $80.58 \pm 0.7$ | $111.3 \pm 0.4$ |
| | 2 | $33.2 \pm 0.3$ | $108.7 \pm 0.9$ |
| | 1 | $16.8 \pm 0.2$ | $24.1 \pm 0.7$ |

Table 6: Comparison of inference time for Decision Diffuser and Consistency Planning on the different datasets with respective default diffusion steps (Ajay et al., 2022).

| Method | Dataset | Inference Time (ms per sample) | Avg. Norm Score |
|---|---|---|---|
| Decision Diffuser | Hopper-m | $1954.3 \pm 5.4$ | $79.1 \pm 3.6$ |
| | Hopper-mr | $1938.5 \pm 3.7$ | $100.5 \pm 0.4$ |
| | Halfcheetah-me | $1920.4 \pm 4.5$ | $89.2 \pm 1.2$ |
| | Walker2d-me | $1899.9 \pm 7.8$ | $107.5 \pm 2.1$ |
| Consistency Planning | Hopper-m | $33.5 \pm 1.8$ | $88.6 \pm 1.1$ |
| | Hopper-mr | $33.7 \pm 1.1$ | $96.9 \pm 1.3$ |
| | Halfcheetah-me | $33.1 \pm 2.1$ | $94.1 \pm 0.9$ |
| | Walker2d-me | $32.9 \pm 0.8$ | $108.5 \pm 0.7$ |

