# OpenReview forum: "Planning with Consistency Models for Model-Based Offline Reinforcement Learning"
_TMLR — Accepted by TMLR_

### Review · Reviewer_d7Jk · 2024-09-14

**Summary Of Contributions:**

This paper presents an approach that combines guided consistency models with the decision diffuser framework. Consistency Planning maintains performance comparable to diffusion model counterparts while significantly improving the computational efficiency of planning. This advancement is particularly crucial for tasks requiring agile motion planning.

**Audience:**

Yes

**Broader Impact Concerns:**

I think there are no related ethical concerns about this paper.

**Claims And Evidence:**

Yes

**Requested Changes:**

Critical changes that would impact recommendation for acceptance:

1. Include experimental results for both the proposed method and comparison algorithms on long-horizon tasks, such as Maze2D.
2. Submit the source code of the algorithm for review. This will allow for a thorough examination of the implementation details and facilitate reproducibility of the results.

Changes that would strengthen the work:

1. Elaborate on any challenges encountered during the implementation of the method, as well as potential issues that might arise when applying this approach to real-world robotic tasks.
2. Provide a comprehensive time efficiency comparison between the proposed method and Consistency-AC.

**Strengths And Weaknesses:**

Strengths:

1. The introduction of consistency models enhances planning speed, which is vital for tasks with high real-time requirements.
2. The authors have effectively demonstrated the efficacy of combining consistency models with the decision diffuser, although this outcome was somewhat predictable.

Weaknesses:

1. There appears to be a performance gap between the proposed method and Consistency-AC. It would be interesting to see how the method performs on long-horizon tasks, such as Maze2D.
2. The time efficiency comparison with Consistency-AC is not clearly presented, which would help bolster confidence in the paper's claims.
3. The paper lacks a discussion on potential implementation challenges. It's worth noting that open-source libraries like CleanDiffuser (https://github.com/CleanDiffuserTeam/CleanDiffuser) could potentially simplify the implementation of the proposed algorithm. However, this observation does not affect my overall decision on the paper.

---

> ### Author Response · Authors · 2024-10-15
> **Response to Reviewer Comments**
>
> Thank you for your detailed and constructive feedback. We appreciate your recognition of the strengths of our approach and your suggestions for improvement.
>
> In response to your comments, we have added experimental results for both our method and Diffuser on the long-horizon task Maze2D, which are included in the revised submission (see Section 5.1, page 8). These results further demonstrate the effectiveness of our method in long-horizon planning scenarios, as requested.
>
> Additionally, we are happy to submit the source code of our algorithm for review. We can provide a anonymous GitHub URL (https://github.com/anonymity738/consistency_planning) to ensure the code remains anonymous while allowing for a thorough examination of the implementation details.
>
> Regarding the suggestions about potential issues that might arise when applying this approach to real-world robotic tasks, we add the concerned discussion in the Section 6.
>
> Regarding the other suggestions about providing a more detailed time efficiency comparison with Consistency-AC, we acknowledge the importance of these points and plan to address them in future revisions.
>
> Thank you again for your valuable feedback and suggestions.

---

### Review · Reviewer_y3Mz · 2024-09-27

**Summary Of Contributions:**

The paper introduces Consistency Planning, a novel approach that integrates consistency models with the Diffusion model framework to improve computational efficiency in model-based reinforcement learning (RL) for sequential decision-making tasks. Diffusion-based models, such as Diffuser and Decision Diffuser, have demonstrated strong performance but suffer from slow inference processes. The proposed method leverages the computational efficiency and speed of consistency models during the inference process, which significantly reduce the number of sampling steps required, without sacrificing the performance  and robust planning of the diffusion-based approaches. The method is validated on D4RL benchmark tasks, showing a 12-fold increase in speed compared to diffusion models, with comparable performance.

**Audience:**

Yes

**Claims And Evidence:**

Yes

**Requested Changes:**

see above.

**Strengths And Weaknesses:**

**Strengths:**
 - The paper effectively addresses a key issue in diffusion-based RL models:  computational inherent inefficiency during inference due to iterative reverse processes. By introducing Consistency Planning, the authors provide a valuable alternative that maintains high performance while drastically speeding up the inference process.
 - The integration of consistency models trained through distillation from pre-trained diffusion models into Decision Diffuser is an interesting concept. The authors clearly articulate the process of using inverse dynamics models to derive actions, showcasing how consistency models can be adapted from diffusion models for planning tasks.
- The use of consistency models significantly reduces the number of steps required during inference, addressing the main challenge of slow sampling in diffusion models. This is a notable improvement, especially in real-time or resource-constrained environments.
- The results on D4RL tasks provide evidence that the proposed method achieves comparable performance with diffusion-based methods while offering over a 12-fold speedup. The experiments compare to model-free approaches and to behaviour cloning and model-based methods, offering a broad evaluation framework.

**Weaknesses:**
 - While the method section is detailed, it might be challenging for readers unfamiliar with diffusion and consistency models to follow. For example, the notation and equations could benefit from more intuition or explanation about how they contribute to the final results. The section on the inverse dynamics model and consistency distillation is mathematically dense, and a clearer breakdown of the concepts for general readers might improve accessibility.
- While Consistency Planning offers a substantial speed improvement, its performance is slightly lower than state-of-the-art models like Diffusion-QL and Consistency-AC (87.9 vs. 85.1). The paper could provide more insights into why this trade-off occurs and under which conditions the slight drop in performance becomes significant.
- The paper focuses on offline RL, but the application of consistency models in online RL settings could be a valuable extension and make the method more broadly applicable. The method's potential limitations or benefits in real-time decision-making tasks are not explored. The authors could discuss the potential challenges or benefits of applying the approach to scenarios where quick, real-time decision-making is critical.
- The integration of multiple techniques (diffusion models, consistency models, inverse dynamics, etc.) might make the proposed method more challenging to implement in practice. More details on how these components interact and their impact on overall system complexity could be helpful for practitioners
- While the performance results are positive, the paper primarily compares its method against diffusion-based baselines. More direct comparisons with other state-of-the-art model-based reinforcement learning approaches or those employing different planning strategies could further substantiate its claims. Additionally, while speedup is highlighted as a core advantage, a deeper dive into the computational costs (e.g., memory consumption, training times) compared to traditional diffusion models could add to the paper’s impact. Therefore I think providing a deeper analysis of the trade-offs between inference speed and performance would strengthen the paper. For example, investigating whether the slight drop in performance in some tasks is a general limitation of consistency models or specific to certain types of environments.
- The future work outlined (e.g., designing better weighting functions, combining with advanced techniques in consistency models) could have been explored more. Even a brief experiment with one of these techniques could demonstrate the potential for further improvements, offering readers a better glimpse of what’s possible

---

> ### Author Response · Authors · 2024-10-15
> **Response to Reviewer Comments**
>
> Thank you for your detailed and constructive feedback. We have made several revisions based on your suggestions to enhance the clarity, completeness, and depth of the paper. Below is a summary of the changes made in response to your specific comments:
>
> 1. Clarifying the method section:
>    In response to your concern about the complexity of the method section, we have added a clearer breakdown of the concepts for the inverse dynamics model and consistency distillation in Section 4. Additionally, we included an ablation study on the inverse dynamics and guidance scale in Section 5.2 to further clarify how these components contribute to the final results.
>
> 2. Discussion on performance trade-offs:
>    Regarding the performance difference compared to state-of-the-art models like Diffusion-QL and Consistency-AC, we have added a discussion in Section 5.1 to explain why our method performs slightly lower in the Half-Cheetah environment and under which conditions this trade-off becomes more significant.
>
> 3. Future work on online RL:
>    While we did not include experiments in an online RL setting, we have outlined the application of our method to online RL as a valuable extension in the future work section (Section 6). We also discuss the potential limitations of our method, particularly in real-time decision-making tasks, in this section.
>
> 4. Clarifying system interactions:
>    To address your comment on the complexity of integrating diffusion models, consistency models, and inverse dynamics, we have updated Figure 1 (the framework of Consistency Planning) and ablation study of inverse dynamics model (see Section 5.2) to better illustrate how these components interact and contribute to the overall system.
>
> 5. Analysis of computational costs and performance trade-offs:
>    Although we did not dive deeply into the computational costs (e.g., memory consumption, training times) compared to traditional diffusion models, we have added discussions about the lower performance in Half-Cheetah in Section 5.1 and potential limitations in Section 6. This helps to clarify the trade-offs between inference speed and performance in specific environments.
>
> 6. Future work and advanced techniques:
>    While we did not conduct an experiment to explore the combination with advanced techniques (e.g., better weighting functions) as part of our future work, we acknowledge the importance of this direction. Unfortunately, we did not obtain overall better results from such attempts, so they were not included in this version.
>
> We hope these revisions address your concerns and improve the overall quality of the paper. Thank you again for your valuable feedback.

---

### Review · Reviewer_mDYs · 2024-10-01

**Summary Of Contributions:**

The paper proposes the use of consistency models for model-based offline RL, which allows for faster inference, which is a problem with diffusion models due to their iterative sampling process.

**Audience:**

Yes

**Broader Impact Concerns:**

I don't have any concerns.

**Claims And Evidence:**

Yes

**Requested Changes:**

- Please add more details to experiments (see Weaknesses point 2 above).
- Please define the metrics you have used.
- Please provide full details to reproduce the experiments (hyperparameters, architectures and solver choices).
- Please also discuss sensitivity to your choices of hyperparameter choices, training hardware and times.
- Please annotate the algorithms steps with comments so that it is easy to follow.
- Bold the best numbers in Table 1 so that it is easy to parse.

**Strengths And Weaknesses:**

Strengths:
- The speed improvements due to a consistency model are good.

Weaknesses:
- Given that speed of inference is one of the main highlighted contributions, it does not make sense that the paper only presents Figure 2 and concludes that there is a 12x speedup.
- The paper appears to be very incomplete - the experiments are sparse, the authors don't discuss the choice of their D4RL domains (when would the method work, when would it not work), don't provide ablations to their (implicit) choices of hyperparameters.

---

> ### Author Response · Authors · 2024-10-15
> **Response to Reviewer Comments**
>
> Thank you for your thoughtful and detailed feedback. We have made several revisions in response to your suggestions to improve the completeness and clarity of the paper. Below is a summary of the changes made based on your specific requests:
>
> 1. Additional details in experiments (addressing Weaknesses point 2):
>    We have included an ablation study on inverse dynamics and guidance scale in Section 5.2 to provide more insight into the method's performance. Additionally, we discuss when the method works well and when it might exhibit lower performance, as presented in Section 5.1.
>
> 2. Definition of metrics:
>    In response to the request for clearer metric definitions, we have added the following sentence in Section 5.1: “For performance metric, we use normalized average returns as defined by Fu et al. (2020) for all tasks.”
>
> 3. Reproducibility details:
>    To ensure reproducibility, we have added more detailed descriptions of our architecture, solver choices, and hyperparameters in the appendix. This provides a clearer picture for reproducing our experiments.
>
> 4. Discussion of hyperparameter sensitivity and training hardware/times:
>    We have conducted additional experiments on how the guidance scale affects performance across three Hopper environments in Section 5.2. A discussion of these results has been added to the revised version, ensuring a more comprehensive understanding of the sensitivity to hyperparameter choices.
>
> 5. Annotated algorithm steps:
>    We have added annotations to the pseudo-code in the algorithm steps, making it easier to follow and understand the key components of the implementation.
>
> 6. Bolded best numbers in Table 1:
>    In Table 1, we have bolded the best performance numbers to make the table easier to parse, as requested.
>
> We hope these revisions address your concerns and enhance the quality of the paper. Thank you again for your valuable feedback.

---

> > ### Comment · Reviewer_mDYs · 2024-10-21
> >
> > Thank you for addressing my comments. A few follow-ups:
> >
> > 1. Can you be specific about the RL problem spaces the method addresses (discrete states/actions etc) in the problem setting so that it is obvious to the reader.
> > 2. I still don't see several experimental details, like dataset size (it would be good to mention it for completeness even if these are standard datasets). How much data do you expect to do well, given this is an offline RL setting?
> > 3. One minor typo: in Sec 5.2, you have referenced Table 2 (for inverse dynamics), when you should be referencing Table 3.

---

> ### Author Response · Authors · 2024-10-24
> **Response to Reviewer Comments**
>
> Thank you for your follow-up questions and for highlighting these important points. We have made several updates to the manuscript to address your concerns:
>
> 1. Problem Spaces for RL Setting:
> We have clarified in Section 5.1 that the experimental environments used in our study are characterized by continuous state spaces and continuous action spaces. This specification should provide readers with a clearer understanding of the problem setting addressed by our method.
>
> 2. Dataset Details and Characteristics:
> We have added details about the dataset sizes to the appendix for completeness. Additionally, the appendix now includes a description of the three distinct data types used in our experiments for the Hopper, HalfCheetah, and Walker environments.
>
> 3. Discussion on Data Requirements in Offline RL:
> In Section 5.1, we have added a discussion on how action optimality and dataset size influence model performance.
>
> 4. Typo Error:
> The typo in Section 5.2, where we incorrectly referenced Table 2 for inverse dynamics, has been corrected to reference Table 3.
>
> We hope these revisions and explanations address your concerns and provide further clarity. Thank you again for your valuable feedback.

---

### Decision · Action_Editor_KHyF · 2024-11-06

**Recommendation:** Accept as is

**Comment:**

All three reviewers recommended acceptance and I agree with their judgement.

**Audience:**

Enforcing consistency within sequential decision making is a relevant task that should be of interest to a significant subset of the TMLR audience.

**Claims And Evidence:**

While one reviewer suggested that further experimental evaluation could strengthen the paper even more, all three agreed that the paper presents a novel contribution to an interesting problem with sufficient evidence.